# Effect of Modifiers on Self-Healing and Rheological Properties of Asphalt Binder

**DOI:** 10.3390/ma17133304

**Published:** 2024-07-04

**Authors:** Qipeng Zhu, Cuiran Liu, Yanhong Wang, Yanzhen Su, Mingxia Li

**Affiliations:** School of Civil Engineering, Luoyang Institute of Science and Technology, No.90 Wangcheng Avenue, Luolong District, Luoyang 471023, China; 15156918149@163.com (Q.Z.); 200900600387@lit.edu.cn (C.L.); 200900600394@lit.edu (Y.W.); 200902201792@lit.edu.cn (Y.S.)

**Keywords:** asphalt, self-healing, alkane chains, plastics, polymer, wax

## Abstract

The effects of four modifiers were studied to compare their roles in the self-healing ability of asphalt binder: elemental sulfur, with a known plasticizing effect; wax, containing long alkane chains (>C50) with a known crystallizing capability; a plastic oil, with short alkane chains (<C20); and a combination of the wax and plastic oil. The results indicated that the self-healing performance of asphalt binder was increased by each of the four modifiers at 25 °C. Sulfur provided the greatest improvement in self-healing capacity, followed by plastic oil, wax combined with plastic oil, and then wax. This was attributed to sulfur’s plasticizing effect, which is conducive to the self-diffusion of asphalt binder molecules. But at 40 °C, the wax, plastic oil, and wax + plastic oil modifiers weakened the self-healing capacity of asphalt binder, especially the wax modifier. In terms of percent recoverable strain, the asphalt binder modified by wax (long alkane chains) showed the highest percent recovery. The wax-modified asphalt binder also had the highest complex shear modulus compared to the other three modified binders and control binder. And its phase angle was lower than that of plastic oil-, sulfur-modified asphalt binder, and control binder.

## 1. Introduction 

The use of liquid modifiers of asphalt binder has attracted more and more interest for two purposes: to enhance the workability of asphalt mixtures by enabling a reduction in mixing and compaction temperatures and to rejuvenate aged asphalt binder, to allow for more use of reclaimed asphalt pavement (RAP) and reclaimed asphalt shingles (RASs) as a means of conserving resources and recycling.

Researchers have more and more focused on utilizing waste plastic in asphalt pavement to improve service performance, especially with the increasing stock of waste plastic [1]. There is evidence which showed that the usage of HDPE and LDPE in asphalt can increase the rutting resistance [2]. And PE particles influenced the performance of asphalt pavement by changing its different sizes and contents in the mixture [3]. Other types of waste plastics, such as polypropylene (PP), ethylene-vinyl acetate (EVA), acrylonitrile-butadienestyrene (ABS), polyethylene terephthalate (PET), and polyvinyl chloride (PVC) were also utilized for asphalt binder modifiers in some papers [4,5,6]. In general, there are many studies on waste plastic particles applied in asphalt binders and mixtures, but only a few researchers focus on plastic oil used in asphalt. 

Some wax-based additives have also been found to be effective as rejuvenators [7]. Other rejuvenators include various other forms of alkanes and aromatics. A number of rejuvenators have been proposed, such as petroleum-based aromatic extracts, distilled tall oil, and organic oil [8]. Ahmae et al. indicated that the rutting evaluation from rheological analyses of aged and blended binders indicated that the characteristics of blended binders were dependent on the bio-based rejuvenator (JCO) [9]. Adding 3–4% waste cooking oil rejuvenated aged asphalt binder (Performance Grade 40/50) to a condition where its physical and rheological properties closely resembled those of the original asphalt binder (Performance Grade 80/100) [10]. Some studies have found that waste cooking oil is one of the rejuvenating agents that can possibly improve the properties of aged asphalt binder to a level similar to those of virgin asphalt binder [11,12].

There is an interesting phenomenon observed in asphalt binder called self-healing, which counteracts damage from cracking. Some microcracks are self-healed under a load-free state, resulting in a longer asphalt pavement service life [13,14]. Cohesive healing within the asphalt binder was found as one of the forms of self-healing behavior [15,16]. Intrinsic healing happens when there is a strength obtained by a wetted crack interface along with time. Bhasin et al. pointed out a model that revealed the healing mechanism in asphalt binder [17]. During the wetting process, the surfaces of the crack contact each other. There is a density gradient perpendicular to the plane of crack. Due to Brownian motion, some molecules are restored to the low-density area by self-diffusion. The rejuvenator encapsulation method is also a promising automatic self-healing approach for asphalt pavements [18]. And the self-healing efficiency increased with the increase in microcapsule content [19]. Through rejuvenator diffusion into the asphalt binder, the composite of the microcapsules and asphalt binder was softened. And the aged asphalt binder had a multifaceted recovery ability due to the microcapsules containing rejuvenator [20]. After leaking into the microcracks and with the help of capillarity, the rejuvenator filled the cracks with a movement speed mainly determined by the volume of the microcapsules in the asphalt binder [21]. Some rejuvenators were analyzed and compared based on the behavior of mixtures including reclaimed asphalt pavement (RAP). The results indicated that using a rejuvenator additive substantially benefited the self-healing capability of RAP mixtures [22].

Among the properties affected are the self-healing characteristics of asphalt binder, which can change with the additives that are introduced. While some additives improve the self-healing capacity, others negatively impact self-healing. For instance, it has been documented that the presence of wax-based additives hinders the self-healing of asphalt binder at the surface level [23]. This paper examines the effects of several rejuvenators on the self-healing characteristics of asphalt binder at the bulk level and examines the molecular mechanisms that affect asphalt binder’s self-healing capacity. The findings can provide new exploration ideas for improving the self-healing ability of asphalt binder while addressing the problem of plastic pollution.

## 2. Materials and Methods

This study used a PG 64-22 asphalt binder from HollyFrontier Corporation in Arizona (Dallas, TX, USA). Wax (long alkane chains) was acquired from Sasol Chemicals USA LLC. (Houston, TX, USA) Sulfur was acquired from Fisher Scientific™ (Waltham, MA, USA). By the hydrothermal liquefaction (HTL) process, the plastic oil (short alkane chains) was obtained from waste plastics. HTL experiments were conducted equipped with a magnetic stirrer, 4843-controller, and a jacketed heater. The testing temperature was 350 °C, and the container was a 250 mL stainless steel bench-top batch reactor (Parr Instrument Company, Moline, IL, USA). Due to the reactants’ expansion during the heating process, the whole volume of the system was set to a maximum of 125 mL to ensure the smooth progress of the experiment. The plastic oil was produced at 20% solid loading (25 g dry weight) during all HTL experiments. Each modifier was added to the base binder at 3 wt.% (by the weight of the base binder) and then mixed for 5 min. 

### 2.1. Dynamic Shear Rheometer Test

An Anton Paar MCR 302 dynamic shear rheometer (Graz, Austria) was used in this study to investigate the rheological properties of the asphalt binders. The testing temperature was 25 °C; it was controlled through an air bath. The shear strain level was 0.1%, and the frequency was 10 Hz. After testing, the complex shear modulus and phase angle were obtained. The complex shear modulus |G*| and phase angle δ of asphalt binder containing different modifiers were also evaluated. Three replicates were used for each type of sample.

### 2.2. Multiple-Stress Creep Recovery Test

To determine the effect of modifiers on the elastic response in an asphalt binder, the multiple-stress creep recovery test (MSCR) was conducted under shear creep and recovery at two stress levels at a specified temperature. The specified temperature used was 40 °C. The samples’ preparation and testing equipment were in accordance with ASTM D7405 [24], which requires a 25 mm parallel-plate geometry and a 1 mm gap setting. There were four replicates for each kind of sample test. Every sample was subjected to a constant loading stress level for 1 s, then allowed to a recovery phase for 9 s. Twenty creep and recovery cycles at 0.1 kPa creep stress were run and followed by ten creep and recovery cycles at 3.2 kPa creep stress. Based on the collected strain values, the average non-recoverable creep compliance $J_{nr}$ and the average percent recovery *R* with different stresses (0.1 kPa and 3.2 kPa) were calculated. According to each of the last 10 cycles at a creep stress of 0.1 kPa, the adjusted strain value at the end of the recovery portion of every cycle was obtained using Equation (1). The strain value at the end of the recovery portion (that is, after 10.0 s) of each cycle shall be denoted as ε_r_.
(1)$$\varepsilon_{r}\left(0.1/3.2,\;N\right)=\frac{\left(\varepsilon_{1}-\varepsilon_{10}\right)\times 100}{\varepsilon_{1}}\;\;\;\;\;\;\;\;\mathrm{for}\;N\;=1\mathrm{to}10$$

Then, using the results of the strain values, we calculated $J_{nr}$ and *R* at the 0.1 and 3.2 kPa stress levels using Equations (2)–(5).
(2)$$J_{nr}\left(0.1,\;N\right)=\frac{\varepsilon_{10}}{0.1}\;\;\;\;\;\;\;\mathrm{for}\;N=11\;to\;20\;\;$$
(3)$$J_{nr}\left(3.2,\;N\right)=\frac{\varepsilon_{10}}{3.2}\;\;\;\;\;\;\;\mathrm{for}\;N=11\;to\;20\;\;\;$$
(4)$$R_{0.1}=\frac{SUM\left(\varepsilon_{r}\left(0.1,\;N\right)\right)}{10}\;\mathrm{for}\;N=11\;to\;20\;\;$$
(5)$$R_{3.2}=\frac{SUM\left(\varepsilon_{r}\left(3.2,\;N\right)\right)}{10}\;\mathrm{for}\;N=11\;to\;20$$

### 2.3. Healing Test

Following ASTM-D7552 [25], a rheometry test was adopted using time sweep (loading–rest–loading) to analyze the self-healing performance of asphalt binders. The samples were 8 mm in diameter and 2 mm thick which were tested at 25 °C, a strain of 5%, and a frequency of 10 Hz. There were three replicates for each sample. During testing, the first loading was carried out on each sample until its complex shear modulus (G*) was reduced to 50% of the initial value. The loading was then stopped, and the sample was given a rest period of 900 s. Then, the second loading was carried out. The healing index (HI) was then calculated using Equation (6), following prior work [26,27,28].
(6)$$HI=\frac{G_{b}-G_{a}}{G_{0}-G_{a}}\;$$
whereG_0_ is the initial dynamic shear modulus during the first loading;G_a_ is the dynamic shear modulus at the end of the first loading;G_b_ is the dynamic shear modulus before the second loading.

## 3. Results and Discussion

### 3.1. Rheological Properties of Binders

The phase angle and dynamic shear modulus |G*| of the control asphalt binder and four kinds of modified asphalt binders are shown in Figure 1a.

The effects of modifiers on the asphalt binders’ |G*| were assessed by calculating the |G*| increase compared with control asphalt binder. It indicated that the wax-modified asphalt binder had a |G*| value 109.8% higher than that of the control asphalt binder and also higher than the other modified asphalt binders. The |G*| values of the control asphalt binder and wax-, plastic oil-, wax–plastic oil-, and sulfur-modified asphalt binders were 7.96 × 10^5^, 1.67 × 10^6^, 4.87 × 10^5^, 1.08 × 10^6^, and 6.25 × 10^5^, respectively. The significant difference in the complex modulus of asphalt binder modified with wax is attributed to the crystallizing capability of the wax’s long alkane chains. This is mainly because the alkanes in the wax used in this study are C50, and these alkanes are more prone to crystallization at typical service temperatures.

Figure 1b shows the phase angles of the control asphalt binder and four modified binders. The phase angles of the control asphalt binder and wax-, plastic oil-, wax–plastic oil-, and sulfur-modified asphalt binders were 66.06, 60.11, 67.34, 60.01, and 66.825, respectively. The wax-modified binder had a phase angle of 5.95° which was lower than that of the control asphalt binder. This shows that the induction of wax increased the elastic portion of asphalt binder. The phase angles of plastic oil- and sulfur-modified asphalt binders were respectively 1.28° and 0.765°. They were all larger than that of the control asphalt binder. Additionally, comparing the four kinds of modifiers, it can be seen that wax or wax combined with plastic oil had more significant negative effects on the asphalt binder’s phase angle, and plastic oil had the most significant positive effect on the asphalt binder’s phase angle, followed by sulfur.

### 3.2. Elastic Response Property of Binders

Figure 2 shows the results of the multiple-stress creep recovery (MSCR) test.

The non-recoverable creep compliance ($J_{nr}$) and deformation recovery percent (R) of modified asphalt binders for the stress levels of 0.1 kPa and 3.2 kPa were calculated using Equations (1)–(5). The asphalt binder modified with sulfur showed the largest $J_{nr}$ value in magnitude, indicating the highest susceptibility to rutting. In contrast, the wax-modified asphalt binder showed the lowest $J_{nr}$, indicating the greatest resistance to rutting. At the creep loading level of 0.1 kPa, the $J_{nr}$ values for the control asphalt binder and wax-, plastic oil-, wax–plastic oil-, and sulfur-modified asphalt binders were 0.0943, 0.0274, 0.0995, 0.0757, and 0.1084/1/kPa, respectively. At the creep loading level of 3.2 kPa, the $J_{nr}$ values for the control asphalt binder and wax-, plastic oil-, wax–plastic oil-, and sulfur-modified asphalt binders were 0.0967, 0.0330, 0.1038, 0.0959, and 0.1121/1/kPa, respectively. Compared to the $J_{nr}$ values for wax- and plastic oil-modified asphalt binders, at the creep loading level of 0.1 kPa, the $J_{nr}$ value of the wax–plastic oil-modified asphalt binder was respectively higher by 0.0483 and lower by 0.0238/1/kPa; at the creep loading level of 3.2 kPa, it was respectively higher by 0.0629 and lower by 0.0079/1/kPa. Compared to the $J_{nr}$ values for sulfur- and plastic oil-modified asphalt binders, at the creep loading level of 0.1 kPa, the $J_{nr}$ value of the wax-modified asphalt binder was respectively lower by 0.081 and 0.0721/1/kPa; at the creep loading level of 3.2 kPa, it was respectively lower by 0.0791 and 0.0708/1/kPa. It can be seen that a higher creep loading level resulted in a higher $J_{nr}$ for all asphalt binders.

Comparing R values, the asphalt binder modified with sulfur showed the lowest R value, and the wax-modified asphalt binder showed the highest R value. At the creep loading level of 0.1 kPa, the values of the percent recovery for the control asphalt binder and wax-, plastic oil-, wax–plastic oil-, and sulfur- modified asphalt binders were 22.15%, 43.31%, 24.59%, 38.85%, and 22.13%, respectively. At the creep loading level of 3.2 kPa, the values of the percent recovery for the control asphalt binder and wax-, plastic oil-, wax–plastic oil-, and sulfur-modified asphalt binders were 20.01%, 34.10%, 21.47%, 26.02%, and 19.56%, respectively. Comparing the percent recovery of different modified asphalt binders, the wax modifier had a significantly improved effect on asphalt binder. At the creep loading level of 0.1 kPa, the R value of the wax-modified asphalt binder was respectively 21.18% and 18.72% higher than the R values of sulfur- and plastic oil-modified asphalt binders. At the creep loading level of 3.2 kPa, the R value of the wax-modified asphalt binder was respectively 14.54% and 12.63% higher. Compared to the R values for wax- and plastic oil-modified asphalt binders, at the creep loading level of 0.1 kPa, the R value of the wax–plastic oil-modified asphalt binder was respectively lower by 4.46% and higher by 14.26%; at the creep loading level of 3.2 kPa, it was respectively lower by 8.08% and higher by 4.55%. In terms of percent recoverable strain, wax (with long alkane chains) showed the highest percent recovery.

### 3.3. Self-Healing Property of Binders

Figure 3 shows the healing index calculated according to Equation (6). Among the four modified asphalt binders, the sulfur-modified asphalt binder had the highest healing index, which means its healing property was the best. This is mainly because the phenolic compounds in asphalt binder can work as an activator for sulfur interactions within the asphalt binder. This in turn indicates that the effect of sulfur can be more noticeable in a bituminous matrix [29]. The healing indexes of the control asphalt binder and wax-, plastic oil-, wax–plastic oil-, and sulfur-modified asphalt binders were 23.35%, 30.46%, 32.66%, 31.65%, and 35.27%, respectively. Compared to the healing index of the control asphalt binder, the healing indexes of wax-, plastic oil-, wax–plastic oil-, and sulfur-modified asphalt binders were higher by 7.11%, 9.31%, 8.3%, and 11.92%, respectively. Among the four modifiers, sulfur had the most significant improvement in the healing performance of asphalt binder. This was attributed to sulfur’s plasticizing effect, which facilitates the self-diffusion of asphalt molecules. Although the wax-modified asphalt binder had the lowest $J_{nr}$ value and highest R value, the molecules in wax are more prone to crystallization, making the wax-modified asphalt binder stiffer than other modified asphalt binders. The increased stiffness will reduce the diffusion of asphalt binder molecules across a crack boundary and thereby lower the healing index. Additionally, the presence of wax will accelerate the formation of “bee” structures, which would hinder crack healing [23]. This is why the healing index of the wax-modified asphalt binder was lower than that of the sulfur-modified asphalt binder. Also due to this reason, introducing wax into the plastic oil-modified asphalt binder decreased the healing performance of the asphalt binder.

The findings also showed that the wax modifier improved the self-healing performance of the asphalt binder compared to the control asphalt binder. This was mainly due to their being investigated at different levels: surface and bulk. Albert [23] and Huang [30] investigated and evaluated the effect of wax on the self-healing property of asphalt binder using surface and bulk properties, respectively. The wax modifier’s improvement in healing performance aligns with previous research that associated modifiers such as waxes containing linear alkanes with improved healing properties [31]. Additionally, according to Bhasin’s healing mechanism model, the surface free energy of the asphalt binder dictates the initial phase of intrinsic healing, and for the subsequent time-dependent intrinsic healing, it is dictated by the self-diffusion of asphalt molecules across the crack interface [32]. So, the healing that occurs on the surface of a crack is only a part of the healing process. Compared to the control asphalt binder, each of the four modifiers showed a positive effect on the healing index of asphalt binder.

The healing property of the modified asphalt binders was also tested at 40 °C to analyze the influence of temperature on healing performance. Figure 4 shows the testing results. All the healing index values at 40 °C were higher than those at 25 °C.

This is due to the rheological property becoming stronger when the temperature increased. Therefore, it became more effective to heal the crack area. But only the healing index of the sulfur-modified asphalt binder was still better than that of the control asphalt, while the healing index values of the other three modified asphalt binders were lower. At a higher temperature, it means that the other three modifiers besides sulfur all weaken the healing performance. Comparing the results of wax-, plastic oil-, and wax–plastic oil-modified asphalt binders, it was concluded that with an increase in temperature, the negative effect of wax became more significant. This is because the ‘bee’ structures [33], which are believed to be wrinkled lamellar thin films of crystalline wax, lead to nanoscale surface roughness, which results in more micro-voids following the initial crack that weaken the ‘healed’ material with mild heating [23]. This is why the asphalt binders with wax modifier were shown to have a worse healing index under a higher temperature. After introducing the plastic oil into the wax-modified asphalt binder, the healing index decreased further.

## 4. Conclusions

Since self-healing is influenced by the diffusion of asphalt binder molecules across a crack boundary, modifiers with a plasticizing or crystallizing effect could significantly affect the self-healing of asphalt binders. The effect of four modifiers on the self-healing characteristics of asphalt binders was studied. The following are the conclusions which are drawn from the evaluation of the resulting modified bitumen. And in the future, the effect of modifiers on the performance of asphalt mixtures could be a good point to explore.
Adding each of the modifiers to asphalt binder increased its healing capacity at 25 °C. Among the four modifiers, sulfur provided the greatest improvement in healing capacity, followed by plastic oil, wax with plastic oil, and wax. Sulfur’s performance was attributed to sulfur’s plasticizing effect, which facilitates the self-diffusion of asphalt binder molecules.Sulfur modifier can improve the healing performance, but the other three modifiers (especially wax) weakened the healing performance of asphalt binder at 40 °C. In terms of percent recoverable strain, wax showed the highest percent, while the other additives did not show significant changes relevant to control asphalt binder.The wax-modified asphalt binder had the highest complex shear modulus compared to the other three modified binders and control binder. And its phase angle was lower than that of plastic oil-, sulfur-modified asphalt binder, and control binder. The latter improvement is attributed to wax crystallization, which aligns with the findings of prior studies.The significantly higher complex modulus of the asphalt binder modified with wax compared to modifying with plastic oil is attributed to the crystallizing ability of the wax’s alkane molecules. The alkanes in plastic oil range from C13 to C20 molecules; the alkanes in wax are C50 molecules, which are more prone to crystallization at typical service temperatures.


## Figures and Tables

**Figure 1 materials-17-03304-f001:**
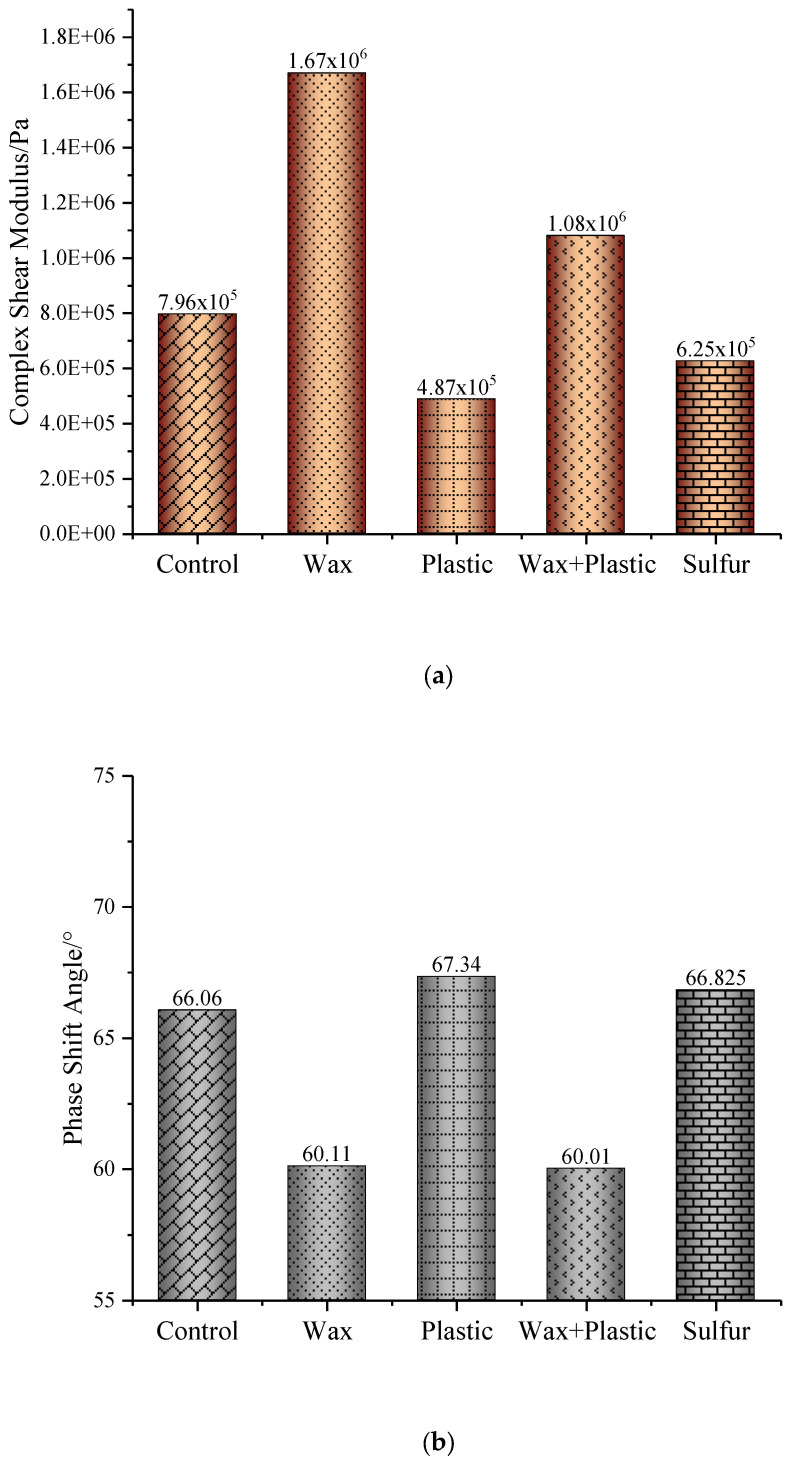
(**a**) Dynamic shear modulus and (**b**) phase angle of control asphalt binder and modified asphalt binders.

**Figure 2 materials-17-03304-f002:**
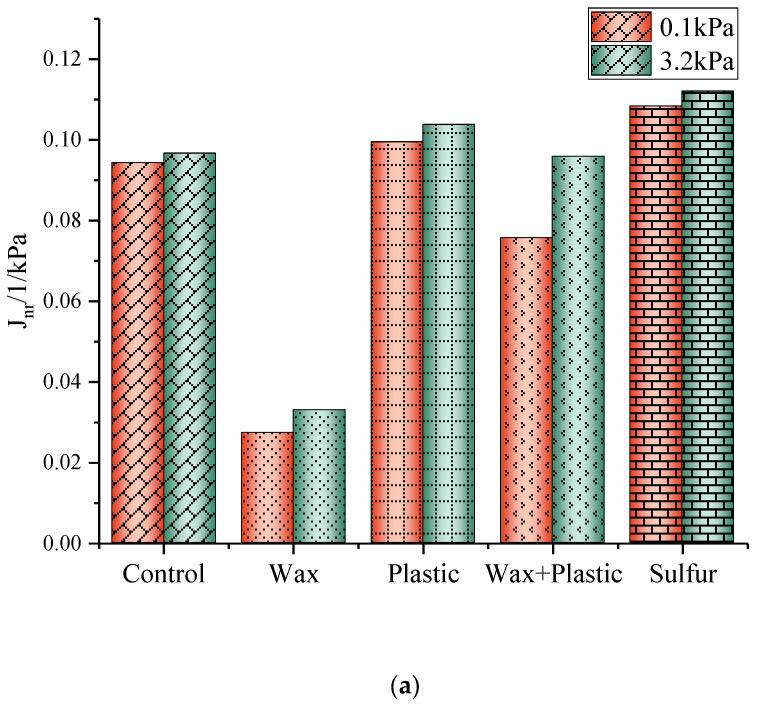
(**a**) Non-recoverable creep compliance and (**b**) deformation recovery percent of control asphalt binder and modified asphalt binders.

**Figure 3 materials-17-03304-f003:**
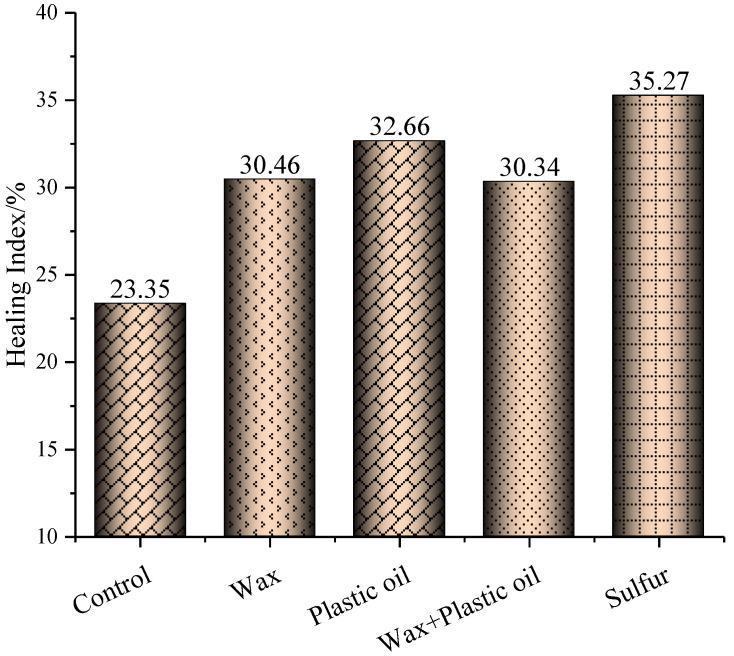
Healing index of control asphalt binder and modified asphalt binders.

**Figure 4 materials-17-03304-f004:**
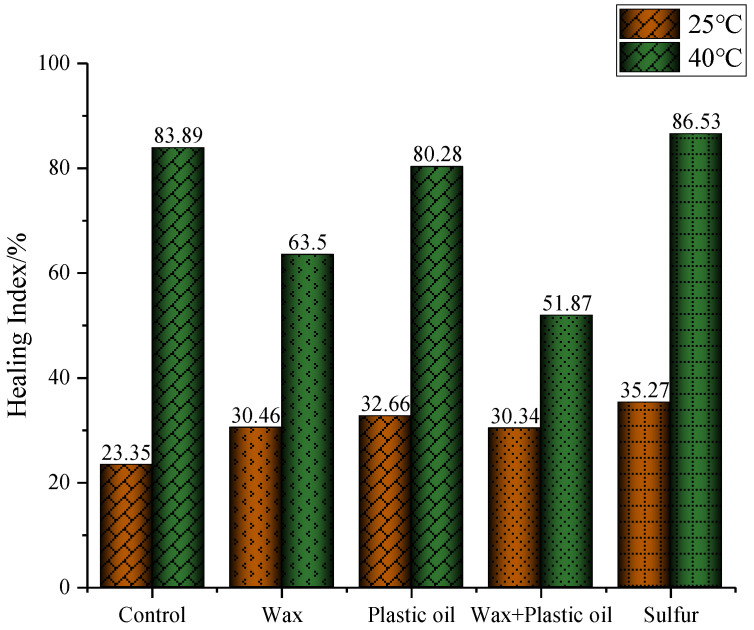
Healing index at two temperatures for control asphalt binder and modified asphalt binders.

## Data Availability

The data that support the findings of this study are available on re-quest from the corresponding author.

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
