# Peer review of "Effect of Modifiers on Self-Healing and Rheological Properties of Asphalt Binder"

_materials, 2024, doi:10.3390/ma17133304_

Round 1

Reviewer 1 Report

Comments and Suggestions for Authors

- The title of this paper covers an extensive area. It would be a good idea to limit the scope.

- The literature review is poor.

- You don`t need to describe in detail MSCR.

- For G* and δ, what range of temperature and frequencies have you used?

- Figure 1 doesn`t say too much without knowing frequency

- What is "Kpa?"

-  you should make your figures look more professional. The patterns and colors don`t look too good.

- Self-healing should be related to asphalt mixtures. I can`t see any test with mixtures.

- Conclusions are too long. Look like another discussion.

Overall, the paper doesn`t contribute much to current knowledge.

Comments on the Quality of English Language

English must be improved. Here is an example of the first sentence of the introduction: "The use of liquid modifiers of asphalt binder has appealed more and more interests due to for two purposes." 

Reviewer 2 Report

Comments and Suggestions for Authors

The article under review investigates the impact of four different modifiers on the self-healing properties of asphalt binders. These modifiers include elemental sulfur, wax with long alkane chains, plastic oil with short alkane chains, and a combination of wax and plastic oil. The study provides a comparative analysis of these modifiers at different temperatures (25 °C and 40 °C) to determine their efficacy in enhancing the self-healing performance of asphalt binder.

The study's findings are significant for the field of materials science, particularly in the development and optimization of asphalt binders for improved durability and longevity. At an elevated temperature of 40 °C, the wax, plastic oil, and their combination adversely affected the self-healing ability of the asphalt binder, with wax having the most pronounced negative impact. This suggests that while these modifiers may enhance performance at lower temperatures, their efficacy diminishes or reverses at higher temperatures. The article provides a comprehensive analysis of how different chemical modifiers affect the self-healing capabilities of asphalt binders. The results suggest that sulfur is the most effective modifier at lower temperatures, enhancing the material's self-healing capacity significantly. Conversely, the performance of wax, plastic oil, and their combination varies with temperature, necessitating careful consideration in practical applications. 

In my opinion, the article can be accepted for publication after minor corrections. It should be noted, however, that the authors should show more scientific novelty, as this research is rather basic (compared with, for example, Di Wang, Andrea Baliello, Lily Poulikakos, Kamilla Vasconcelos, Muhammad Rafiq Kakar, Gaspare Giancontieri, Emiliano Pasquini, Laurent Porot, Marjan Tušar, Chiara Riccardi, Marco Pasetto, Davide Lo Presti, Augusto Cannone Falchetto, Rheological properties of asphalt binder modified with waste polyethylene: An interlaboratory research from the RILEM TC WMR, Resources, Conservation and Recycling, Volume 186, 2022, 106564,ISSN 0921-449, https://doi.org/10.1016/j.resconrec.2022.106564).

Reviewer 3 Report

Comments and Suggestions for Authors

The paper investigates the impact of four different modifiers—elemental sulfur, wax, plastic oil, and a combination of wax and plastic oil—on the self-healing and rheological properties of asphalt binder. The study concludes that sulfur provides the greatest improvement in self-healing capacity at 25°C, followed by plastic oil, wax combined with plastic oil, and wax. At 40°C, most modifiers, particularly wax, reduce the self-healing capacity. The rheological tests indicate that wax-modified asphalt binder has the highest complex shear modulus and the highest percent recovery in terms of recoverable strain.  The manuscript needs improvements before it can be recommended for publication. Please see below further comments for consideration:

-Include numerical results in the abstract to provide a quantitative overview of the findings. Specific data points, such as the percentage increase in self-healing capacity and complex shear modulus values, should be mentioned. -The introduction lacks depth and a comprehensive literature review. Expand on the background and significance of the study by including more references to recent and relevant studies. -Clearly state the research gap and the novelty of the study. Explain why the selected modifiers are important and what specific problems they address in the context of asphalt binders. -Remove unnecessary details about the sources of materials (e.g., HollyFrontier Corporation, Sasol Chemicals USA LLC, Fisher Scientific™). Instead, focus on the characteristics and specifications of the materials used. - Provide bibliographic references for the equipment and methods used, such as the hydrothermal liquefaction process and the dynamic shear rheometer test, to support the methodology. - Ensure all testing procedures are clearly described with sufficient detail to allow replication of the study. Include details about sample preparation, test conditions, and measurement techniques. -  The results section should include a detailed statistical analysis. Report standard deviations, confidence intervals, and p-values to validate the findings. -  Address the potential limitations of the study and suggest areas for future research to further investigate the observed phenomena. -  Include specific numerical results in the conclusions to substantiate the claims made. For example, mention the exact percentage improvements in self-healing capacity and complex shear modulus.

Comments on the Quality of English Language

 Extensive editing of English language required

Reviewer 4 Report

Comments and Suggestions for Authors

The paper materials-3035351 "Effect of Modifiers on Self-Healing and Rheological Properties of Asphalt Binder" investigates the effectiveness of four modifiers (or rejuvenators?) to improve self-healing properties of asphalt.

I have some questions for the authors:

1. the English language should be improved

2. There are some typing errors in the manuscript

3. Multi-figures do not comply with the journal standard

4. the abstract is too vague and presents only qualitative results (e.g., lower, higher. Since the study is comparative, this approach cannot be accepted.

5. A big question: did the authors investigate rejuvenators or modifiers? The review literature focuses on rejuvenators, the rest of the paper and the title deals with modifiers. They are not the same.

6. Not all variables of Equations 1 to 5 are defined. Is SUM the summation symbol?

7. The Conclusion section does not present the results.

8. Are there further developments for this study?

Comments on the Quality of English Language

the English language should be improved

Reviewer 5 Report

Comments and Suggestions for Authors

The article "Effect of Modifiers on Self-Healing and Rheological Properties of Asphalt Binder" investigates the effects of sulfur, wax, plastic oil, and a combination of wax and plastic oil on asphalt binder's self-healing ability. At 25°C, all modifiers improved self-healing, with sulfur being the most effective due to its plasticizing effect. At 40°C, however, wax, plastic oil, and their combination reduced self-healing, especially wax. Wax-modified binder showed the highest strain recovery, complex shear modulus, and lowest phase angle, indicating superior rheological properties despite its diminished high-temperature self-healing performance.

 COMMENTS

 The percentage match 37% of iThenticate report is high 

https://susy.mdpi.com/ithenticate/manuscript/get_report/9fe3f3c86034af545c81292c7783153b

Please try to reduce it.

In Figs 1, 2, etc., and in many places in the text:  kPa    [NOT Kpa !!!]

The presented results are limited to specific cases. I have concerns since in my opinion this is a limited work and is not so strong to be published in this journal.

Can the authors point out the originality of their article?

Some Recommendations for Future Work could be added at the end of the Conclusions or as a separate section after Conclusions.

Comments on the Quality of English Language

Some editing of the English language is required.

Round 2

Reviewer 3 Report

Comments and Suggestions for Authors

the paper can be accepted

Comments on the Quality of English Language

Minor editing of English language required

Reviewer 4 Report

Comments and Suggestions for Authors

the authors revised the paper according to the suggestions. the paper can be accepted-

Comments on the Quality of English Language

Minor editing of English language required

Reviewer 5 Report

Comments and Suggestions for Authors

The revised paper can now be accepted for publication.

Comments on the Quality of English Language

Minor editing of the English language is required which will be checked anyway by the journal before proofs.